# The Transcriptional and Translational Landscape of Plant Adaptation to Low Temperatures

**DOI:** 10.3390/ijms26178604

**Published:** 2025-09-04

**Authors:** Aleksandra V. Suhorukova, Olga S. Pavlenko, Denis S. Sobolev, Ilya S. Demyanchuk, Valery N. Popov, Alexander A. Tyurin

**Affiliations:** Timiryazev Institute of Plant Physiology, Russian Academy of Sciences, 127276 Moscow, Russia; sualsha@yandex.ru (A.V.S.); helliga.p@gmail.com (O.S.P.); denissoboleww@gmail.com (D.S.S.); ilya.demyanchuk.2000@yandex.ru (I.S.D.); vnpopov@mail.ru (V.N.P.)

**Keywords:** *Solanum lycopersicum*, tomato, temperature stress, WGCNA, GO-enrichment

## Abstract

One of the unresolved questions in stress-response biology is how plants coordinate expression levels between the response and adaptation. In this work, we proposed a two-level analysis that examines both transcriptional and translational profiles of *Solanum lycopersicum* under conditions of short-term cold stress, hardening, and their combination. By combining polysome profiling and total transcriptome analysis, we revealed that expression under cold stress is not a simple linear process but a structurally distinct system with two coordinated regulation centres. Hardening triggers a strong transcriptional program focused on biogenesis, light signalling, and structural adaptations. In contrast, acute stress prompts selective translation of metabolic and defence proteins without prior transcriptional increase. Modular analysis (WGCNA) showed little overlap between transcriptional and translational networks, indicating functional differences between regulation levels. This work demonstrates that the cold response involves a strategic reallocation of resources between transcription and translation based on the type of signal. It bridges basic biology and applied breeding, providing targets promising for improving plant stress tolerance and advancing bioengineering of adaptive agriculture.

## 1. Introduction

Throughout the long evolutionary process, plants have faced many different environmental challenges, one of which is the deleterious effects of low temperatures. Despite the long-term trend of increasing average temperature, plants are regularly affected by low-temperature stress, which is one of the key abiotic factors that inhibit their growth, development and geographical spreading [1]. Spring frosts are the most destructive effects caused by changing weather conditions. In this case, the preservation of metabolic activity, as well as the survival of the plant as a whole when exposed to low temperatures, implies the initiation of a number of complex adaptation mechanisms that entail various kinds of biochemical, physiological, and structural changes [2,3]. Generally, plant response to cold affects many aspects of plant life, forming collectively a concerted chain of reactions at molecular, cellular, and organismal levels, which ultimately aims to provide plant resistance to cold stress [4].

According to the current model of low-temperature perception by plants, some of the first transformations to which they undergo as a result of cold exposure are a decrease in the fluidity of cell membranes and, as a consequence, conformational changes in membrane proteins, including receptor proteins. Excitation of cold receptors together with depolymerisation of the cytoskeleton of plant cells leads to the initiation of complex signalling cascades accompanied by the involvement of secondary messengers, namely calcium ions and reactive oxygen species (ROS), which transmit external signals inside the cell [5]. In the course of signal transduction, there is regulation of the activity of various kinases that perform both the processes of temperature signal perception and propagation and activation of cold response gene expression, including transcription factors (TFs). One of these factors is CBF (C-repeat binding factor), whose expression significantly increases in response to cold stress. CBF expression is positively regulated in the presence of ICE1 (Inducer of CBF Expression 1), in contrast to MYB15, which has the opposite effect. CBF directly binds to cis-elements in the promoters of COR (cold-regulated) genes, inducing their expression with the subsequent formation of various osmotic and antifreeze proteins responsible for the protection and subsequent stabilisation of cellular structures, preventing the formation of intracellular ice crystals, and eliminating the effects of dehydration. Due to the excessive formation of ROS (reactive oxygen species) accompanying cold shock, antioxidant defence mechanisms are also triggered in plant cells to reduce oxidative stress [5,6]. In addition, lipids, as the main part of cell membrane structure, undergo changes that ensure membrane fluidity due to the action of desaturase enzymes aimed at increasing the content of unsaturated fatty acids in their composition [7]. This concept of plant reactions in response to cold stress gives only a partial idea of the adaptation mechanism of their survival under unfavourable environmental conditions.

For a complete understanding of the interaction of the processes occurring during the response of plants to low-temperature stress, it is necessary to pay special attention to the comparison of gene expression regulation systems at the transcription and translation stages with dynamical cellular resolution. This approach is intended to expand the understanding of the mechanisms of protein synthesis adaptation and modification processes under stress conditions. Cold stress alters the transcription pattern of many genes; some of them are inhibited, as, for example, it happens with genes responsible for energy production during photosynthesis, presumably to prevent photooxidative damage to the photosynthetic apparatus. On the contrary, another set of genes is expressed to a higher degree, as their products represent defence proteins and enzymes of the antioxidant system necessary for plant adaptation. One of the regulatory tools continuously changing during cold exposure are transcription factors, which are able to coordinate the adaptation process through the interaction of a complex regulatory network. Meanwhile, it has been reported that a substantial fraction of cold-inducible genes (approximately 25%) encode TFs [8]. To date, many different TF families are known; some of them are CBF/DREB, MYB, bHLH, WRKY, NAC, bZIP, etc., which at the molecular level are directly involved in the regulation of plant adaptation mechanisms. These transcriptional factors are able to recognise and bind to specific motifs located in the promoters of stress-responsive genes, providing a fine regulation of their expression and consequently shaping the plant response [9,10].

However, a high level of transcripts of certain genes does not guarantee a high level of the corresponding proteins, because before proceeding to translation, mRNA undergoes a number of post-transcriptional changes, such as splicing, polyadenylation, and capping of pre-mRNA, which regulate its stability and translational availability. MicroRNAs (miRNAs) are considered to be among the key post-transcriptional regulators of expression, capable of leading to the degradation of mRNAs with which they are complementary. Moreover, many miRNA target genes encode transcription factors, some of which are responsible for plant reactions to cold stress. Also, during post-transcriptional changes, pre-mRNA undergoes alternative splicing (AS), which produces different mRNA variants, increasing the diversity of translated proteins [11]. At the translational level, stress conditions induce plants to selectively translate key proteins critical for survival, while the synthesis of other proteins is reduced or ceases altogether, thus optimising the energy losses of the cell. Translational regulation of pre-existing mRNAs is a rapid alternative way to control gene expression [12]. Also, one of the key aspects of the cellular response is the temporal shift in activation of transcription and translation processes. The occurrence of stress entails a rapid change in the transcript levels of various genes, but their translation may be delayed to varying degrees. This flexible mechanism of temporal coordination of transcription and translation processes allows the synthesis of the most essential proteins for survival to occur first, thereby freeing up cellular resources for higher priority targets. An approach based on the integration of transcription and translation data has the potential to expand the understanding of how plants overcome environmental challenges and ensure long-term tolerance to low temperatures.

The aim of this study was primarily to demonstrate significant functional differences between expression profiles at the transcriptional and translational stages in plants exposed to a stress factor (namely, low temperatures).

## 2. Results

### 2.1. Assessment of the Level of Plant Stress Due to Low Temperatures

After the experimental tomato (*Solanum lycopersicum*) plants were exposed to the specified temperature conditions, we had to confirm their (the plants’) physiological status. For this purpose, we applied widely used tests for biochemical markers of low-temperature stress: sugars accumulation, electrolyte output, and malonic dialdehyde accumulation.

### 2.2. Sequencing of mRNA Libraries

All mRNA libraries were sequenced in two technical repetitions on the Illumina 6000 platform. At least 30 M (Appendix A) reads per sample in each direction (PE) were obtained (detailed information about readings is accessible in GEOsample.csv table in Appendix A Section). The length of the reads was 150 bp. Phred quality score of the reads filtered for further analysis was 30 or more. Raw reads as well as preprocessing results (read counts received with kallisto) are deposited in GEO (GSE282483).

### 2.3. Analysis of Primary Sequencing Results Using Principal Component Analysis (PCA)

We applied the principal component analysis (PCA) to form a first impression of the data and to reduce the dimensionality of the data. The principal component distribution (Figure 1) shows that all experimental samples fall into two groups: samples reflecting transcriptional dynamics and translational dynamics. In our opinion, the compact subclustering of transcripts from hardened plants (Hardening and Combo) separately from control and stressed plants can be explained quite simply—hardening is a time-consuming process, unlike the short-term stress exposure used in this study. That is, the model plants under conditions of low positive temperatures had time to change the expression profile at the transcriptome level, whereas plants exposed to short-term low temperatures did not have such an opportunity. In general, transcripts from total RNA (matched transcription) also formed a significantly more compact cluster than translational samples (polysome fraction). This pattern may be partly explained by the greater lability of translation in terms of regulation.

### 2.4. Differential Expression Analysis

Differential expression analyses were performed for all comparisons, identifying three groups of genes: Up—genes with increased expression (logFC > 1, *p*-value cutoff = 0.05), Down—genes with decreased expression (logFC < −1, *p*-value cutoff = 0.05), and Stable—genes with stable transcription levels (−1 <= logFC <= 1, *p*-value cutoff = 0.05). After searching for differentially expressed genes using the edgeR package, the initial results were presented as UpSet plots (Figure 2), where the height of the bars in the plot reflects the size of the intersection of the DEGs sets for the corresponding comparison variants. Despite the rather general nature of the data presented, some striking features can be identified.

For the Up group, i.e., for genes with increased expression, we can note the size of the primary DEGs sets ranging from 3000 to 4000 genes, with the Combo_vs_Hargening and Stress_vs_Control comparisons falling out of this pattern, which can be explained in the same way as the distribution of PCA data—stress (in the presented experiment) is a short-term effect (2 h), which may not be sufficient to significantly alter the transcriptional profile. The Combo_vs_Stress, Combo_vs_Control, and Hardening_vs_Control samples show a curious similarity in the composition of DEGs, both when compared pairwise and jointly. If we keep in mind that the Stress sample is not significantly different from the Control sample and the Combo sample is not significantly different from the Hardening sample, it is not difficult to find an explanation for these results.

The distribution of DEGs compositions with reduced expression (Down group) almost completely qualitatively coincides with the results of the Up group, which does not require additional explanation.

Special attention should be paid to the group of genes with stable expression. In this case, the sizes of gene sets for individual comparisons reflect not differentially expressed genes but genes common to both comparisons. The Combo_vs_Stress, Combo_vs_Stress, Combo_vs_Control, Hardening_vs_Control, and Hardening_vs_Control samples demonstrate a certain similarity in the composition of DEGs (623 genes).

However, simply comparing samples of DEGs is not the most informative approach, primarily due to the different sizes of these samples. To offset the impact of differences in the size of DEGs sets, we calculated Jaccard coefficients and presented them as heat maps (Figure 3) for a more relevant comparison of differential expression analysis data. With this representation, groups with common gene subsets remain the same as when considering the primary data while gaining quantitative characterisation.

### 2.5. Differential Translation Analysis

While transcriptomic data (reflecting the expression profile at the transcriptional stage) are suitable for direct comparison, polysome sequencing results (indirectly reflecting translation intensity) cannot be compared directly with each other. This is primarily due to the direct influence of transcript start amounts on the number of ribosome-bound transcripts. Since the number of translated transcripts largely depends on the initial number of mRNA molecules of a given type, normalization of poly/total is required—the so-called translational ratio—how effectively molecules of a particular mRNA type enter the translation process. Also, because transcript length directly affects the number of bound ribosomes (although this relationship may be indirect and rather unobvious), it makes sense to normalise differential expression data to transcript length for within-sample comparisons, but this is not necessary for between-sample comparisons. 

Translational efficiency—the ratio of the number of RFPs (of a given gene/data isoform) to the total number of reads mapped to the coding region of a given gene/data isoform.

Change in translational efficacy—the ratio (or difference in the case of logarithmic values) of the translational efficacy values for the two samples being compared [13].

Translational ratio—the ratio of the number of transcripts translated (of a given gene/data isoform) to the total number of transcripts (of a given gene/data isoform) [14].

Changes in translational ratio—the ratio (or difference in the case of logarithmic values) of the translational ratio values for the two samples being compared.

Thus, to study the dynamics of translation between samples, we calculated the translational rate for each experimental sample as follows:$$I_{i}=logFC\frac{poly}{total}={log}_{2}\frac{poly}{total}$$
And already then the samples were compared with each other according to the following parameter:$$\Delta TR=I_{i}-I_{j}$$

We also analysed differential translation for all comparison variants, identifying three groups of genes: Up—genes with increased translation level, Down—genes with decreased translation, and genes with stable translation level. When examining the resulting UpSet plots (Figure 4), it can be concluded that in contrast to the differential expression data, where the Stress and Control samples had minor differences, the differential translation data show significantly larger differences, which can be explained by a higher rate of translation regulation as opposed to transcription.

For both Up and Down, the largest sizes of primary DTGs sets can be observed in the Stress_vs_Hardening comparison group (2812 and 4033, respectively). At the same time, if we take into account that the Combo sample, in contrast to Stress, has insignificant differences with the Hardening and Control samples, it can be concluded that the hardening method contributes significantly to plant resistance to low temperatures.

It is noteworthy that for the Stable group there is a set of 235 genes whose stable expression level is maintained for all experimental conditions, which is consistent with the Jaccard coefficients for the corresponding group (Figure 5), where the degree of similarity in pairwise comparisons ranges from 12% to 28%. It can be assumed that this gene pool is responsible for the functioning of basic aspects of plant cell viability and retains its translational activity regardless of stress or hardening conditions. However, analyses of gene ontologies did not reveal notable features of GO term enrichment.

### 2.6. Term Enrichment Analysis of Gene Ontologies

#### 2.6.1. KEGG

Up: pathways associated with ribosomal activity and regulation of protein synthesis (KEGG:03010, 03050, 03008) are predominant in the transcription step, particularly pronounced in Stress_T_vs_Hardening_T, Combo_T_vs_Control_T, and Hardening_T_vs_Control_T comparisons. This reflects the activation of genetic programmes that prepare the cell for stress. In contrast, the translation step is dominated by metabolic pathways (KEGG:01100, 01110) related to energy metabolism, biosynthesis of secondary metabolites, and carbohydrate metabolism. This is particularly expressed under Combo_vs_Stress, Combo_vs_Control, and Hardening_vs_Control conditions. A comparison of expression levels within the experimental groups showed that under stress conditions, transcription and translation are regulated independently: active transcription of ribosomal components is combined with metabolic compensation at the level of translation. In hardened plants, a stable combination of regulatory activity at the level of transcription and metabolic rearrangement in translation was observed. Under combined conditions (Combo), transcriptional responses are attenuated, whereas translation remains active, which may indicate the formation of stress memory and a switch of regulation to the post-transcriptional level. Thus, transcription reflects the preparatory phase of the response, while translation provides rapid metabolic adaptation. This emphasises the importance of integrated analysis of both expression levels for understanding the mechanisms of plant cold tolerance.

Stable: the KEGG:03010 ribosome-compared Stress_T_vs_Hardening_T, Combo_T_vs_Control_T, Combo_T_vs_Stress_T, and Hardening_T_vs_Control_T pathway is the most stable in transcription. This reflects the stable activity of the protein synthesis system. The KEGG:01200, 01230, 01100, and 01110 pathways associated with basic metabolism and degradation of substances are also repeated. At the level of translation, stability of metabolic processes prevails. KEGG:01230 is regularly active in several comparisons (Hardening_vs_Control, Combo_vs_Hardening, Stress_vs_Control), indicating maintenance of catabolic activity. Separate stable signals are observed at KEGG:03015, 00250, and 00330 (ribosome assembly, amino acid metabolism), whereas the ribosomal pathway (KEGG:03010) is conserved only in Combo_vs_Control and Combo_vs_Stress. The comparison of levels shows that transcription ensures stable expression of components of the protein machinery, while translation ensures stability of metabolic processes. The highest concordance is observed in the Combo group, which may reflect the physiological stability formed after hardening. Thus, stable genes represent a functional core that maintains basic cellular processes independent of external conditions.

Down: analysis of genes with reduced expression revealed that pathways related to general metabolism (KEGG:01100) and biosynthesis of secondary metabolites (KEGG:01110) are most pronouncedly repressed. These changes are particularly characteristic at the level of translation, where in most comparisons—including Stress_vs_Hardening, Combo_vs_Hardening, and Stress_vs_Control—the number of annotated genes in these pathways exceeds 70. Simultaneously, a number of transcriptional comparisons, such as Hardening_T_vs_Control_T, Combo_T_vs_Control_T, and Combo_T_vs_Stress_T, also show decreased activity of these same metabolic cascades as well as associated carbohydrate and amino acid metabolism pathways (00520, 00195, 00630, 00710, etc.). At the level of transcription, suppression of ribosomal and regulatory pathways—KEGG:03010, 03050, 03008—was prominent, especially in the Stress_T_vs_Hardening_T comparison, which may reflect inhibition of synthesis of components of the translational machinery under conditions of transient or repeated stress. However, ribosomal pathways were suppressed less frequently and less significantly during the translation step, indicating that the main protein synthesising activity was retained even with a general decrease in metabolism. Thus, the reduced expression encompasses both key metabolic processes and individual elements of the translation system. This indicates the transition of cells into the energy-saving mode, in which transcription of most metabolic genes is reduced and translation becomes selective. The most pronounced and complex changes are noted in comparisons involving stressors after hardening, emphasising the significance of the sequence of exposure and the presence of adaptive metabolic rearrangement (Appendix A).

#### 2.6.2. GO:MF

Functional annotation of genes by GO:MF category revealed systematic differences in molecular functions depending on the direction of expression change. The genes with increased expression are characterised by the predominance of functions providing translational response, regulation of protein metabolism, and energy-dependent interactions. In particular, activities related to translation factors (GO:0008135), rRNA binding (GO:0019843), translation initiators (GO:0003743), and GTP- and ATP-binding proteins are enriched. These patterns are particularly represented when comparing stressed and hardened plants, reflecting enhanced translational activity and mobilisation of the protein synthesising apparatus as a key adaptive strategy. The group with unchanged expression exhibits predominantly general molecular functions lacking strict context specificity. Prominent among these are functions of binding and non-specific catalytic activity, including binding profiles and general-purpose enzymatic activities that show neither stress nor metabolic orientation. This spectrum of function appears to correspond to a basic molecular pool that is not subject to dynamic regulation in response to changes in conditions. Genes with reduced expression are predominantly associated with metabolic and signalling functions requiring coenzyme support and ion transport. Enriched activities include metal and ion binding (including calmodulin-dependent), functions related to biotin and iron–sulfur clusters, and redox enzyme activities. In particular, suppression of the activities of NAD(P)H-dependent reductases, flavin-binding enzymes, and carboxylases may indicate systemic inhibition of certain energy and anabolic fluxes under stress conditions. Such functional shifts are particularly noticeable when comparing stressed plants with controls and indicate a prioritised shutdown of part of the metabolic and transduction cascades. Thus, the spectrum of molecular functions strictly correlates with the directionality of the transcriptional response: stress is associated with activation of protein synthesising and translational-regulatory functions, hardening is associated with rearrangement of enzymatic and coenzyme-dependent pathways, and stable functions remain the background of constant activity (Appendix A).

#### 2.6.3. GO:CC

The functional distribution of transcripts and translationally active mRNAs across cellular components (GO:CC) revealed distinct differences in the spatial organisation of the response to cold stress, hardening and their combination at the cell level. At the transcriptional level, genes with reduced expression were predominantly localised in chloroplast and photosynthetic structures, including photosystems I and II, thylakoid membranes, and the plastid envelope, reflecting systemic suppression of light-dependent processes under stress and, in part, in hardened and combined variants. At the same time, suppression of mitochondrial and ribosomal components was observed, especially in the Stress_T_vs_Hardening_T comparison, where structures related to protein transport (vesicles, EPR, and Golgi complex compartments) are also involved. This indicates a broader impact of stress on the energy and protein synthetic infrastructure of cells. In contrast, genes with increased transcriptional activity showed enrichment in terms related to ribosomes, proteasomes, peptidase complexes, and structures involved in protein degradation and translation regulation (e.g., preribosome, small-subunit processome). This is particularly pronounced in the Combo_T_vs_Control_T and Hardening_T_vs_Control_T comparisons. Variants involving stress (Stress_T_vs_Hardening_T) also activated chloroplast photosystems, indicating a possible recovery of photosynthetic function under some conditions. At the level of translation, trends were generally maintained but were more selective. Increased expression was predominantly associated with ribosomal subunits and proteasome complexes in the Stress_vs_Hardening, Combo_vs_Hardening, and Hardening_vs_Control comparisons, emphasising the activation of rapid protein regulation and degradation systems under conditions of recovery and adaptation. Variants showing increased expression of genes encoding chloroplast components (Combo_vs_Stress, Hardening_vs_Control) indicated the possibility of reactivation of photosynthetic processes under certain pretreatment. In turn, the reduced translational activity was concentrated in chloroplast-related structures including photosystems, thylakoids, and plastid membranes, especially in the Stress_vs_Control and Combo_vs_Stress comparisons. This further emphasises the repressive role of cold on photosynthetic activity. Interestingly, genes with unchanged expression at both transcriptional and translational levels were consistently associated with ribosomes and transport vesicles (e.g., ER-to-Golgi vesicles) as well as spliceosomal complexes. This may indicate that a baseline level of transcriptional and translational activity is maintained to ensure cellular homeostasis even under stress. Thus, GO analysis by cell component revealed a multilevel rearrangement of intracellular architecture in response to stress and adaptive interventions (Appendix A).

#### 2.6.4. GO:BP

Under conditions of stress and hardening, decreased expression of genes related to carbohydrate transport (GO:0008643) and pigment metabolism (GO:0042440, GO:0046148) was observed. This may indicate a redistribution of resources under stress conditions aimed at maintaining vital functions. Increased expression of genes involved in the localisation of proteins to organelles (GO:0033365, GO:0072599) indicates activation of processes that ensure proper functioning of cellular structures in response to stress effects. Decreased expression of genes related to amino acid catabolism (GO:0009063) and photosynthetic processes (GO:0009767) may reflect adaptation of metabolic pathways to altered environmental conditions. This allows the plant to efficiently utilise available resources and maintain energy balance. Genes with unchanged expression associated with key metabolic processes, such as amino acid synthesis and photosynthesis regulation, show resistance to stress. This emphasises the importance of maintaining basic cellular functions for plant survival. Thus, a comprehensive analysis of differential gene expression at the transcriptional and translational level under various stresses and hardening has revealed key biological processes involved in plant adaptation. The obtained data may serve as a basis for further studies of molecular mechanisms of plant resistance to unfavourable environmental factors (Appendix A).

#### 2.6.5. WGCNA

To conclude this study, we analysed weighted gene co-expression networks (WGCNA). Following the scheme presented above, data corresponding to transcription and translation were analysed separately. It is worth considering that the initial design of the experiment did not provide for the implementation of WGCNA; therefore, the modules and patterns found should be considered as preliminary and exploratory.

For transcriptional data, 12 modules (groups of genes whose expression shows correlation) were obtained (Appendix A); for data obtained for the translation step, 15 modules were obtained. For each of the steps (transcription and translation), the correlation between module and experimental conditions was examined (summarised data are presented as heat maps in Appendix A). As can be seen for each condition, distinct co-expression modules were identified, differing both in the degree of correlation and in the power of these modules. It is worth noting, however, that in the vast majority of cases, the *p*-value is quite large (values indicated in brackets in the heat maps). Therefore, even if the observed regularities do occur, they cannot be considered statistically reliable, and additional experiments may be required to confirm them. Nevertheless, a certain number of comparisons are statistically significant. For these modules, we performed additional functional enrichment analyses, which showed the following.

For the transcriptional samples, the correlation between the control conditions and the peachpuff module can be considered statistically significant. The correlation coefficient in this case is −0.98 (*p*-value = 0.025), indicating a general decrease in gene expression of this module in control conditions (for other processing conditions, the data are not statistically significant but show a weak positive correlation within 0.2–0.5). This module includes 464 genes. Functionally, these genes are annotated with high-level terms such as cellular component organisation or biogenesis, plastid organisation, cellular component organisation, thylakoid membrane organisation, response to light stimulus, plastid membrane organisation, mitotic cell size control checkpoint signalling, etc. For enrichment with GO:BP terms, in case of KEGG, these are Ribosome, Ribosome biogenesis in eukaryotes, and TCA cycle Krebs cycle (Appendix A).

For the translational samples, 3 modules with *p*-value < 0.05 were identified: peachpuff (395 genes), correlation coefficient −0.98 (*p*-value = 0.02); lightgrey (1979 genes), correlation coefficient −0.99 (*p*-value = 0.014); and darksalmon (1214 genes), correlation coefficient −0.97 (*p*-value = 0.031).

Modules were associated with the following experimental conditions: peachpuff—hardening, lightgrey—a combination of hardening and stress, and darksalmon—stress.

Functional enrichment analysis of the genes of the peachpuff module (for translation) shows the predominance of high-level terms such as translation, macromolecule biosynthetic process, gene expression, biosynthetic process, protein metabolic process, macromolecule metabolic process, negative regulation of gene expression via chromosomal CpG island methylation, protein targeting to ER, etc. For enrichment with GO:BP terms, in the case of KEGG and WP, they are Ribosome, Ribosome biogenesis in eukaryotes, and TCA cycle Krebs cycle. These processes are suppressed in hardened plants.

For the lightgrey module, closely positively correlated with combined effects, genes are annotated with the following terms: positive regulation of ubiquitin protein ligase activity, positive regulation of post-translational protein modification, positive regulation of protein modification by small protein conjugation or removal, positive regulation of transferase activity, regulation of post-translational protein modification, regulation of protein modification by small protein conjugation or removal, positive regulation of protein metabolic process, transcription preinitiation complex assembly, and lipid X metabolic process. For enrichment of KEGG terms: Lipoic acid metabolism, ubiquitin-mediated proteolysis, and amino sugar and nucleotide sugar metabolism.

For the darksalmon module, whose constituent genes are repressed in plants under stress, enrichment is observed for the following terms: transport, acetyl-CoA metabolic process, cellular process, regulation of transport, cellular localisation, regulation of monoatomic ion transmembrane transport, organophosphate biosynthetic process, organophosphate metabolic process, purine-containing compound metabolic process, membrane lipid metabolic process, acyl-CoA metabolic process, purine-containing compound biosynthetic process, nitrogen compound transport, protein transport, lipid metabolic process, regulation of transmembrane transport, positive regulation of translational elongation, positive regulation of translational termination, amide biosynthetic process, lipid biosynthetic process, carboxylic acid metabolic process, regulation of translational elongation, and organic acid biosynthetic process. KEGG term enrichment: Biosynthesis of unsaturated fatty acids and Valine, leucine, and isoleucine biosynthesis.

In addition to identifying and analysing the modules within each step, two groups of modules corresponding to each step were compared. As expected, based on the data described above, only 2 groups of modules show similarities (as expressed using the Jaccard coefficient) greater than 5%. If we look in more detail, the darksalmon modules for transcription and translation show insignificant similarity at the level of 5% (recall that the names do not indicate similarity, commonality, etc., and are chosen by the programme used for the analysis). The translational modules gainsboro and darkgrey and the transcriptional modules coral, brown, and saddlebrown are more related. Excluding statistically insignificant comparisons, the following 3 comparisons were identified: gainsboro (translation) vs. coral (transcription); darkgrey (translation) vs. brown (transcription); and darkgrey (translation) vs. saddlebrown (transcription) (Figure 6 and Appendix A).

Next, functional enrichment analysis (namely GO-enrichment) was performed for common genes of similar modules. The comparison of gainsboro vs. coral demonstrates the predominance of such terms as MAPK cascade, acyl-CoA metabolic process, membrane fusion, negative regulation of abscisic acid-activated signalling pathway, heat acclimation, regulation of intracellular signal transduction, proteasome-mediated ubiquitin-dependent protein catabolic process, energy derivation by oxidation of organic compounds, positive regulation of signalling, proteasomal protein catabolic process, translational initiation, and carbohydrate derivative catabolic process.

The leading terms for darkgrey vs. brown are oxylipin biosynthetic process, glucan catabolic process, photosynthesis, light harvesting, nitrate metabolic process, polysaccharide catabolic process, galacturonan metabolic process, and response to light intensity. For the last comparison, steroid metabolic process, carboxylic acid catabolic process, cellular response to toxic substances, cellular oxidant detoxification, response to jasmonic acid, and response to fatty acids were identified.

## 3. Discussion

The analysis of transcriptomic and translational data allowed us to comprehensively characterise the response of tomato plants to low-temperature exposure and identify key differences in gene expression regulation at these stages of genetic information implementation.

At the transcriptional stage, cold stress caused significant changes in the expression levels of a large number of genes. This is especially characteristic of plants under hardening and combined exposure conditions, where genes associated with ribosome biogenesis, transcription regulation, signalling pathways, and primary metabolism were activated. At the same time, short-term exposure to cold caused less pronounced transcriptional changes, probably due to the insufficient duration of exposure for a complete transcriptome reorganisation. These differences are consistent with the results of PCA and differential expression analysis, where the main contribution to the dispersion is made by the duration of exposure to low temperatures. Functional analysis (KEGG and GO) confirmed differences in the pathways and molecular functions involved at two levels of regulation. It was found that transcriptionally active genes are more often associated with ribosomal components, transcription regulators, and signalling molecules. In contrast, at the translation stage, transcripts associated with metabolism, biosynthesis of secondary metabolites, and catabolic processes predominated. In our opinion, this may reflect adaptation to different aspects of stress exposure in terms of duration. Analysis of the cellular localisation of differentially expressed genes (GO:CC) clarified the spatial organisation of the molecular response of plants to cold stress. The decrease in the transcriptional activity of genes associated with chloroplasts and elements of the photosynthetic apparatus reflects the general suppression of light-dependent processes under low-temperature conditions. In contrast, genes whose products are localised in ribosomes, proteasomes, and other molecular complexes involved in protein synthesis and utilisation showed increased activity at both the transcriptional and translational levels. This indicates the activation of the cellular protein regulation system, which allows damaged proteins to be removed in a timely manner and replaced with new ones that are functionally significant for adaptation to stress. Co-expression analysis (WGCNA) showed that transcriptional and translational co-expression networks form separate but partially overlapping modules reflecting specific cell states under the studied conditions. Considering (Appendix A) top 10 hub genes has shown prevalence of proteins containing F-box(ubiquitination), HD (nucleic acid metabolism, signal transduction, etc.), GTP1/OBG (uncertain functions), NB-ARC (signalling motif, shared by plant resistance gene products) domains and SNF2-related domain (DNA repair and recombination), ABC-transporters (active trans-membrane transport), and AAA-proteins (ATP-hydrolysis energy transfer). Two features are noteworthy: (i) these domains are divided between transcription and translation (and among the translation modules, and there is also a specification regarding the composition of the domains found); (ii) a significant percentage of domains (~30%) remain uncharacterised. However, we were unable to draw detailed and statistically sound conclusions due to the small number of replicates and comparable samples (which is extremely important for this type of analysis [15], so the identified co-expression modules should be considered preliminary and exploratory. At the same time, we encountered a lack of an acceptable tool for analysing co-expression networks at the translation level. In view of this, the composition of gene groups whose expression correlates at both stages (transcription and translation), i.e., a kind of ‘alarm kit’ of the plant organism, remains unclear.

It is also important to highlight two significant limitations that were not taken into account in this study: (i) the resolution of the data obtained at the organ level, i.e., the responses specific to the tissues that make up the leaves of the model plants were not considered, and, moreover, the experiment was not conducted taking into account the cellular resolution, i.e., the strategies used by different cell types to compensate for the effects of adverse factors. In addition, the presented work did not take into account proteomic and metabolomic data (ii). Deterministic–chaotic dynamics (in particular, the “butterfly effect” type) are characteristic not only of the transcription/translation transition, but also of the translation/protein activity, as well as the protein activity/physiological effect of the metabolite. On the other hand, a holistic approach should also be kept in mind, since different cell types can influence each other within the whole plant organism. All this should be taken into account when extrapolating the presented results [16,17].

## 4. Materials and Methods

### 4.1. Experimental Design

Experimental conditions included control (22 °C), hardening (4 °C, 5 days), low-temperature stress (exposure of non-hardened plants at 0 °C for 2 h), and a combined version (exposure of hardened plants at 0 °C for 2 h). Total RNA preparations were used to obtain information on expression levels, and polysome fractions obtained by polysome profiling were used to analyse the involvement of mRNAs in the translation process.

### 4.2. Cultivation of Experimental Plants

Tomato plants (*Solanum lycopersicum* L., selection line YALF) were grown under controlled conditions in a climate chamber (25 ± 1 °C, 16-h photoperiod, 200 µmol/(m^2^s). The temperature did not change during the daily cycle. 28-day-old plants with 5–7 true leaves were used for experiments. The experimental samples were collected at a fixed time—in the middle of the daylight period—8 h after the lights were turned on (except for plants subjected to stress treatment, options: Stress and Combo variants—middle of the daylight period + 2 h of temperature treatment). Model plants were grown in pots with a diameter of 15 cm and a height of 13 cm, on the soil mix “Terra Vita^®^ Living Soil^®^ Special No. 1” (Terra Vita, Kazan, Russia).

### 4.3. Creation of Low-Temperature Stress Conditions

Hardening and damaging (stress) conditions were created for induction of adaptive mechanisms and subsequent comprehensive evaluation of tomato plant resistance to low temperatures. Hardening was carried out in a climatic chamber LIC-240 (“Binder”, Tuttlingen, Germany) at +4 °C, 16 h photoperiod and illumination 200 µmol/(m^2^s) for 5 days. Damaging (stress) conditions were achieved by chilling plants at 0 °C for 2 h. The parameters of exposure regimes were selected in preliminary experiments.

### 4.4. Determination of Cold Stress Markers

The degree of stress load on plants as a result of low temperatures was assessed by two complementary parameters: relative electrolyte leakage (in %) from leaf tissue to the water phase, an indicator reflecting membrane integrity, and accumulation of malonic dialdehyde, a key marker of lipid peroxidation.

Electrolyte leakage (in %) was calculated using the formula: 100 × (conductivity of the tested sample before or after freezing/conductivity of the same sample after boiling). Electrical conductivity of aqueous extracts was determined using an SG7-ELK conductometer (“Mettler Toledo”, Greifensee, Switzerland) [18].

The content of malonic dialdehyde (MDA) was determined by reaction with thiobarbituric acid. A suspension of leaves (300 mg) was homogenised in 5 mL of extraction medium (0.1 M Tris-HCl buffer, pH 7.6, containing 0.35 M NaCI). To 3 mL of homogenate, 2 mL of 0.5% thiobarbituric acid in 20% trichloroacetic acid was added, incubated at 95 °C for 30 min, cooled, filtered, and optical density recorded at 532 nm wavelength [18,19].

Tomato leaf samples (~500 mg) were fixed in boiling 96% ethanol. The tissue was then ground with a mortar and pestle, and sugars were extracted by triple washing with 80% ethanol.

Glucose content was determined using a glucose oxidase reaction-based kit (GAGO-1KT, Sigma-Aldrich, St. Louis, MO, USA).

For quantitative analysis of fructose and sucrose, the method with resorcinol reaction was used [20].

### 4.5. Total RNA Isolation

Total RNA was isolated from the leaves (2nd–4th levels) by phenol–chloroform extraction using ExtractRNA reagent (Evrogen, cat. no. BC032, Moscow, Russia) according to the manufacturer’s instructions. The obtained samples were additionally treated with DNAase I (Thermo Scientific, cat. no. EN0521, Waltham, MA, USA). RNA concentration and purity were assessed spectrophotometrically using a Synergy H1 microplate reader (BioTek, Winooski, VT, USA). RNA integrity was checked by electrophoresis in 1% agarose gel.

### 4.6. Polysome Profiling

Profiling was performed according to the method described by Lecampion et al. [21], including the following steps:Preparation of a sucrose gradient of four layers (50%, 35% and two layers of 20% each);Salt buffer: 400 mM Tris-HCl (pH 8.4), 200 mM KCl, 100 mM MgCl_2_;Polysome buffer: 4× saline buffer supplemented with 5.26 mM EGTA, 0.5% Triton X-100, 50 μg/mL cycloheximide and 50 μg/mL chloramphenicol;Ultracentrifugation at 175,000× *g* for 2 h 45 min at 4 °C;Precipitation of polysome fractions with isopropanol;Evaluation of the optical OD260 using Synergy H1 microplate reader (BioTek, USA).

### 4.7. Sequencing of mRNA Libraries

Sequencing was performed in two technical repeats for each sample, totalling 24 libraries. Preparation of cDNA libraries and sequencing itself was performed by Evrogen (Moscow, Russia).

TruSeq mRNA Stranded kit (Illumina, San Diego, CA, USA) was used for enrichment of poly(A)+ fractions. The cDNA synthesis was performed using random primers. The resulting cDNA libraries were adapted for the Illumina platform.

The quality of the libraries was checked using the Fragment Analyzer system. Quantification was performed by qPCR. After quality control and concentration normalisation, the pooled library pool was sequenced on the Illumina NovaSeq 6000 platform. Paired-end reads of 150 base pairs in length were obtained. FASTQ files were generated using bcl2fastq v2.20 software (Illumina), and the quality assessment format was Phred + 33.

The total number of reads generated was 2,885,042,754. All raw data are deposited in the GEO database under access number GSE282483.

### 4.8. Primary Data Processing

The quality of raw reads was assessed using FastQC software (version 0.12.1) [22]. Preliminary filtering was performed using the fastp utility, including the following:Adapter removal (TruSeq3);Exclusion of short reads (<30 nucleotides);Filtering for low linguistic complexity (<30%);Base correction and default quality filtering.

### 4.9. Classification of Reads

Reads were classified using the kallisto software (version 0.50.0) [23] based on the tomato transcriptome (assembly SL3.1). Kallisto provides quantification of transcripts without the need for additional counting tools. Pseudo-alignment was performed with bootstrapping (100 iterations), with filtered reads as input.

Classification data are also deposited in GEO (GSE282483).

### 4.10. Differential Expression Analysis

Identification of genes with altered expression levels under different conditions was performed using the edgeR library [24]. The quasi-linear generalised logarithmic regression model with negative binomial distribution was used for variance estimation. Statistical hypothesis testing was performed using the F-test.

Classified reads obtained by kallisto analysis served as input data.

To visualise the results we used the upset library [25] in Python (version 3.10.12), as well as built-in visualisation tools from the edgeR package (version 3.14.0).

### 4.11. GO Enrichment Analysis

Functional analysis of annotated genes was performed using the gprofiler2 package in the R environment, which provides access to the tools of the g:Profiler platform [26].

GO analysis was structured into three levels:Experimental Level—Comparison between experimental conditions.Expression Dynamics—Categorisation of genes into groups with increased, decreased or unchanged expression.Representation Level—Identification of over- and under-represented terms.


Filters were used to increase the biological significance of the results as follows:
term_size ≤ 250—Maximum number of child terms;intersection_size ≥ 10—Minimum number of annotated genes for a term.

The filtered results were visualised as clustered heatmaps for further analysis.

### 4.12. Data Processing and Visualisation

Custom scripts in Python and R were used for batch file processing—including renaming, sorting, and other data operations. Anaconda Software (version 25.7.0) Distribution [27] was used as the development environment, and development was performed in the interactive Jupyter Notebook (version 6.5.7) environment [28].

The following libraries were used for data processing and building visualisations:Python:numpy—For working with arrays;scipy—For statistical analysis;pandas—For processing tabular data;seaborn, matplotlib—For plotting graphs and diagrams.R:tidyverse—a set of packages for processing and visual presentation of structured data.

The GNU Parallel utility [29] was used for parallel processing.

### 4.13. Confirmation of Differential Expression Data by qPCR

Genes that demonstrated significant differential expression by transcriptome analysis were selected for validation. Fold change values were divided into six quantiles, from which representative genes present in all four experimental groups were selected: control, acclimation, stress and acclimation + stress.

Primers were designed using Primer3 software. Their specificity and efficiency were tested both in silico and experimentally.

qPCR reactions were performed on a QuantStudio 5 instrument (Thermo Scientific) using the SYBR Green system (Evrogen) according to the manufacturer’s recommendations. Each reaction was performed in three technical repeats for each sample.

Expression levels were normalised for the housekeeping control genes: RPL2, EF-1-alpha, UBI. Relative expression levels were calculated using the ΔΔCt method.

### 4.14. WGCNA

Weighted gene co-expression networks were analysed using the pyWGCNA library, which implements the functionality of the classical WGCNA package (version 1.73) [30] to work in the Python programming environment [31]. The analysis of transcriptional data did not differ from that proposed by the developers: the input data were expression indices presented in TPM.

At the same time, the analysis of correlations between gene expression levels at the translational level is less developed than at the translational level. For this purpose, we used a pre-calculated parameter—translational ratio. Also, the obtained data were normalised so that the expression levels totalled 1 million for each sample. Such an approach is far from ideal, but we were forced to improvise. 

## 5. Conclusions

The presented study provided only a general idea of the functional specialisation of various stages of genetic information implementation. At the same time, functional differences between transcription and translation in relation to genes that change their expression status are already quite clear. As mentioned above, we assume that such diversification may be associated with overcoming various types of stress exposure of varying duration and, possibly, various aspects of complex stress exposure, which at first glance appears to be monolithic.

Nevertheless, the results of the experiments have raised a large pool of questions that outline directions for possible future experimental work:Development of a methodology for analysing gene co-expression networks at the translational level.The possibility of the existence of trans-level co-expression networks (i.e., co-expression networks whose modules extend to both transcription and translation) and means of studying co-expression networks at both the transcription stage and the translation stage simultaneously.Analysis and study of adaptation strategies specific to different types of plant cells. That is, a detailed, if possible, single-cell study of adaptation to low temperatures.

## Figures and Tables

**Figure 1 ijms-26-08604-f001:**
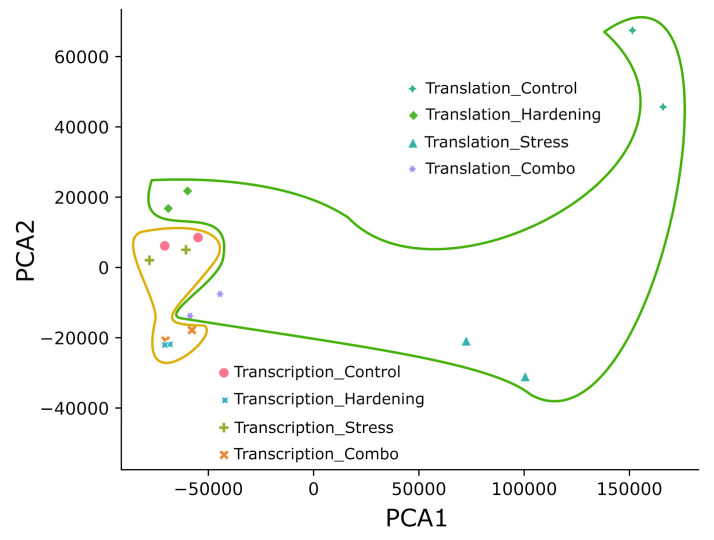
Principal component analysis. The area inside the yellow border is the cluster of all transcriptional data. The area inside the green border is the cluster of all translational data.

**Figure 2 ijms-26-08604-f002:**
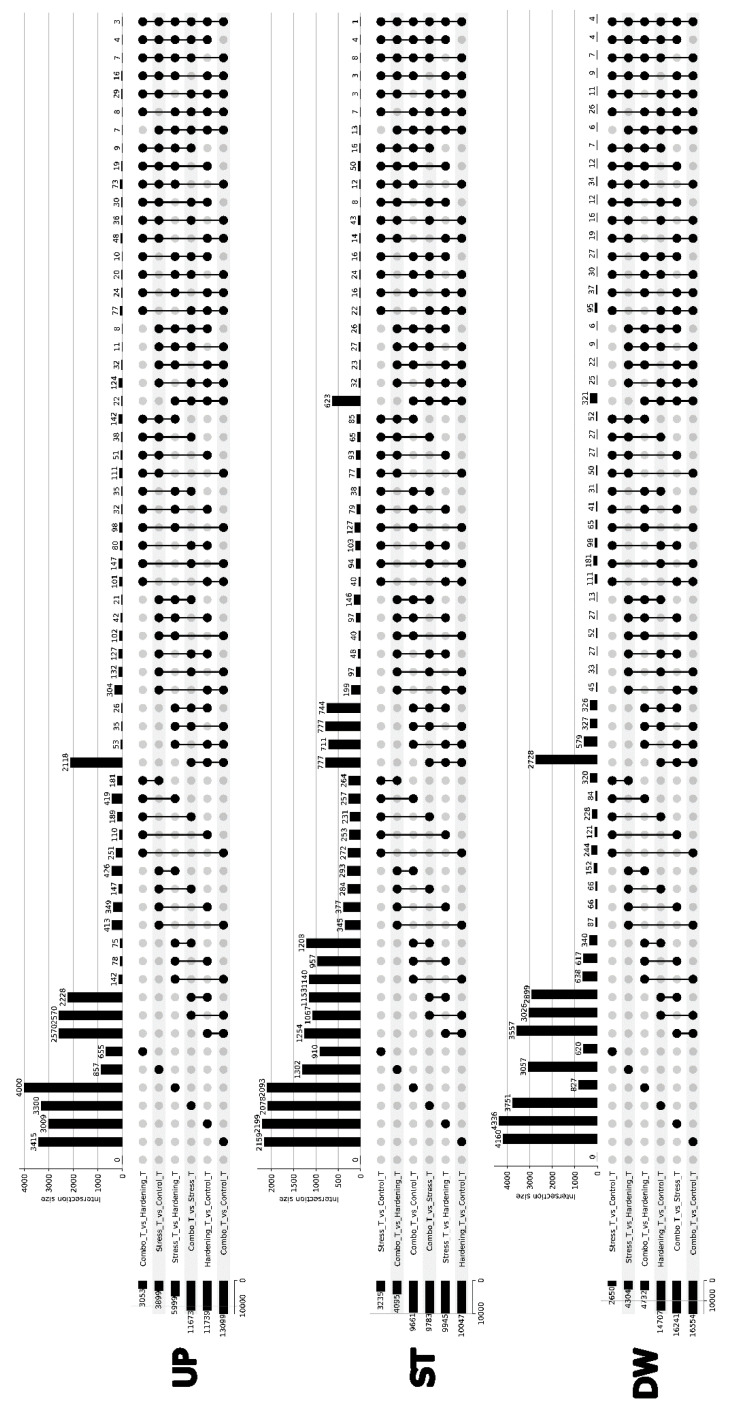
UpSet plot of intersections between differentially expressed gene groups across conditions.

**Figure 3 ijms-26-08604-f003:**
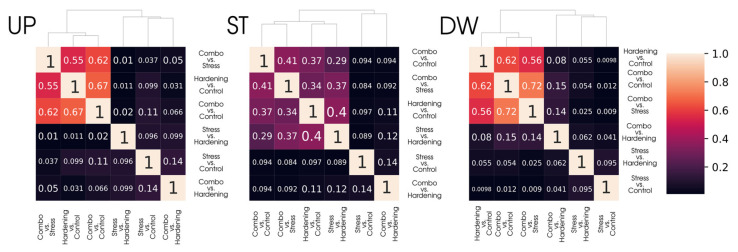
Clustering of differentially expressed gene sets (DEGs) across conditions using the Jaccard Index.

**Figure 4 ijms-26-08604-f004:**
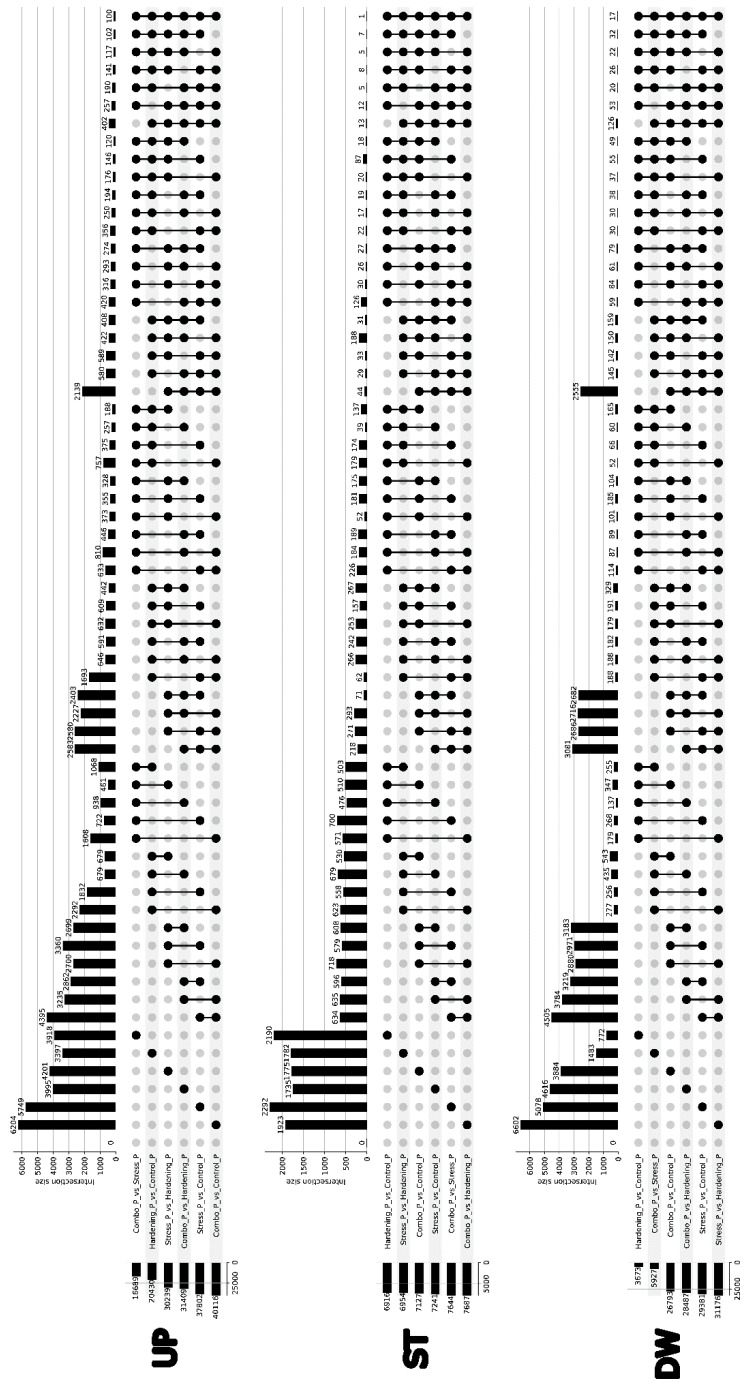
Upset plot showing intersections of genes with differential translational efficiency across conditions.

**Figure 5 ijms-26-08604-f005:**
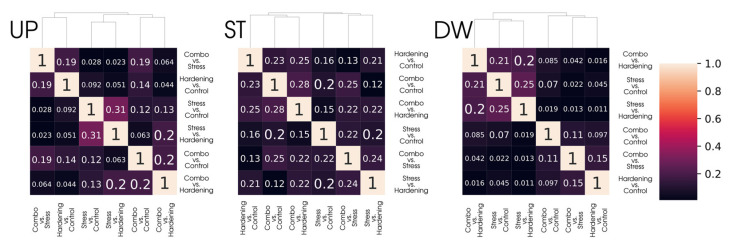
Clustering of differentially translated gene sets across conditions using the Jaccard Index highlighting a stable gene subset.

**Figure 6 ijms-26-08604-f006:**
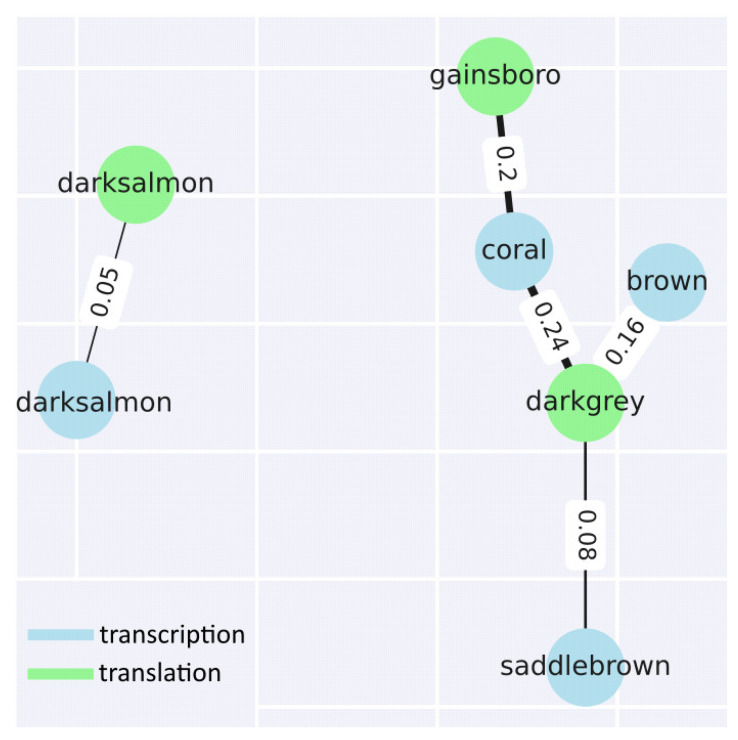
Jaccard-based comparison of transcriptional and translational co-expression modules. Diagram showing the overlap between transcriptional and translational co-expression modules based on the Jaccard index. Most module pairs exhibit low similarity (J < 0.05), except for three significant overlaps: gainsboro (translation) vs. coral (transcription), darkgrey (translation) vs. brown (transcription), and darkgrey (translation) vs. saddlebrown (transcription). These pairs were selected for further GO enrichment analysis. Module names are automatically assigned and do not imply functional similarity.

## Data Availability

All raw data and preprocessed reads are deposited in the GEO database under access number GSE282483.

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
