# Peer review of "The Transcriptional and Translational Landscape of Plant Adaptation to Low Temperatures"

_ijms, 2025, doi:10.3390/ijms26178604_

Round 1
Reviewer 1 Report
Comments and Suggestions for Authors
This manuscript addresses a timely and important question: how plants coordinate transcriptional (slow, preparatory) and translational (fast, selective) responses to cold. By combining total RNA sequencing with polysome profiling and performing differential expression/translation analyses plus WGCNA, the study has potential to reveal mechanistic separation of regulatory “centers” and to identify targets useful in breeding for cold tolerance. However, the conclusions hinge critically on (i) the experimental design (biological replication, fractionation and sequencing depth for polysome vs total RNA), (ii) the robustness and transparency of the normalization/statistical methods used for translational comparisons (polysome data are not the same as Ribo-seq), and (iii) the interpretation of functional enrichments and co-expression modules where many p-values are reported as large. Below, I have provided several questions and suggested fixes, that are needed to ensure scientific integrity, reproducibility, and clarity of this work.
- Section 2.1 Experimental design ( lines 108–114): It is unclear how many biological replicates were used for (a) total RNA and (b) polysome fractions per condition (Control, Hardening, Stress, Combo). The text says “mRNA libraries were sequenced in two technical repetitions” (lines 120–124) , that suggests technical replicates only. I think biological replication is essential to estimate biological variability and to perform differential testing (edgeR etc.). Without adequate biological replicates, p-values and DE calls are unreliable.
-
Section 2.6 Differential translation analysis (≈ lines 187–214, 196–206): The manuscript treats polysome fraction sequencing as a proxy for translation. That is acceptable but requires careful caveats: polysome-association correlates with translation but is also influenced by transcript length, ribosome run-off, and mRNA stability/fractionation artifacts. The methods mention normalizing to starting transcript quantities (lines 190–195) but the exact pipeline is not described.
- The following details should be included:
-Provide a detailed polysome protocol (number of sucrose fractions pooled, OD260 tracing, how polysome vs subpolysome fractions were defined, nuclease treatment or not, controls for ribosomal RNA contamination).
-State how transcript length was accounted for (explicit formula) and whether tximport or other tools were used to import kallisto estimates into edgeR -
You mention preprocessing counts from kallisto (line 123–124) and use edgeR for DE (line 146). kallisto outputs estimated counts (TPM, estimated counts with bootstraps). edgeR expects raw counts and proper handling of kallisto uncertainty (e.g. via tximport). Also the DE thresholds (logFC > 1) are described but p-value / FDR cutoffs are not stated.
-
Up/Down/Stable groups are defined by logFC thresholds only (logFC > 1, < -1, etc.). But classification should combine fold change with statistical significance (adjusted p-value). Also the UpSet plots are shown but numeric overlap metrics (counts per intersection) and Jaccard coefficients (Figure 3) should be accompanied by the sample sizes that generated them.
-
At what circadian time were samples taken? Cold responses and translation are circadian-sensitive. State time of day and ensure all samples were harvested at the same circadian time.
-
At least 30M reads per sample” is that for total RNA and for polysome libraries equally? Provide per-sample read counts in Supplementary Table.
-
There is a wording slip: “The principal component distribution (Figure 1) shows that all experimental samples fall into two groups: samples reflecting transcriptional dynamics and transcriptional dynamics.” (lines 128–131) , likely you meant “transcriptional and translational samples” (fix wording).
-
(lines 187–230): Exactly how were polysome read counts normalized to total RNA counts? Did you compute TE for each gene as (polysome_TPM / total_TPM)? If so, how did you handle low counts? Did you use a statistical framework (Xtail, Babel, RiboDiff) to test significant TE changes? Normalization detail needed.
-
WGCNA typically requires dozens of samples for robust module detection. Please state the number of samples used for WGCNA (transcriptional and translational networks). If sample N is small (e.g., < 12), treat WGCNA results cautiously and state that modules are exploratory.
-
Fix the double phrase in PCA section (“transcriptional dynamics and transcriptional dynamics” (lines ~128–131). There are a few other grammatical slips , run manuscript through careful editing.
-
I suggest to pick ~6 genes that show strong translational regulation but little transcriptional change, and validate by qPCR (total RNA) and polysome fraction qPCR or western blot (protein) to demonstrate true translational regulation (Section 2.6).
- Also, rerun translational efficiency analyses using tools built for TE comparisons (Xtail, Babel, RiboDiff) and compare overlap with your current calls, present this in Supplementary.
- For modules with clear hub genes (WGCNA), provide a short table of top 10 hub genes with annotations and discuss candidate regulators (TFs, RNA-binding proteins) that might mediate the level shift.
Author Response
We are grateful to the referee for reviewing our manuscript and providing constructive criticism. We have revised the language to be more clear and concise, and have included corrections as suggested by the reviewers to enhance the manuscript's clarity and informativeness.
Comments 1: Section 2.1 Experimental design ( lines 108–114): It is unclear how many biological replicates were used for (a) total RNA and (b) polysome fractions per condition (Control, Hardening, Stress, Combo). The text says “mRNA libraries were sequenced in two technical repetitions” (lines 120–124) , that suggests technical replicates only. I think biological replication is essential to estimate biological variability and to perform differential testing (edgeR etc.). Without adequate biological replicates, p-values and DE calls are unreliable.
Response 1: We fully agree that without the necessary number of biological replicates, statistical conclusions become unreliable. In the present study, we used several plants to prepare each library for polysome profiling (due to the need for a large amount of source biological material). Therefore, this cannot be considered a biological repetition in pure sense. However, for sequencing (what was referred to as technical repetitions), different biological samples were used, rather than simply dividing a single source library of total RNA (or polysome fraction).
Comments 2: Section 2.6 Differential translation analysis (≈ lines 187–214, 196–206): The manuscript treats polysome fraction sequencing as a proxy for translation. That is acceptable but requires careful caveats: polysome-association correlates with translation but is also influenced by transcript length, ribosome run-off, and mRNA stability/fractionation artifacts. The methods mention normalizing to starting transcript quantities (lines 190–195) but the exact pipeline is not described.
Response 2: The original description of the data processing process is indeed not clearly described in the text of the manuscript. The appropriate clarifications have been made.
Comments 3: The following details should be included:
-
Provide a detailed polysome protocol (number of sucrose fractions pooled, OD260 tracing, how polysome vs subpolysome fractions were defined, nuclease treatment or not, controls for ribosomal RNA contamination).
-
State how transcript length was accounted for (explicit formula) and whether tximport or other tools were used to import kallisto estimates into edgeR
Response 3: The polysome profiling protocol was specified in the text of the manuscript. The Synergy H1 microplate reader spectrophotometer (BioTek, USA) was used to estimate the optical density. We did not use tximport to import kallisto data; edgeR has a built-in function for this (https://www.bioconductor.org/packages/devel/bioc/vignettes/edgeR/inst/doc/edgeRUsersGuide.pdf , page 13.) We did not make direct corrections for the transcript length, since all comparisons were made between samples, not within them.
Comments 4: You mention preprocessing counts from kallisto (line 123–124) and use edgeR for DE (line 146). kallisto outputs estimated counts (TPM, estimated counts with bootstraps). edgeR expects raw counts and proper handling of kallisto uncertainty (e.g. via tximport). Also the DE thresholds (logFC > 1) are described but p-value / FDR cutoffs are not stated.
Response 4: Added threshold p-values to the text of the article.
Comments 5: Up/Down/Stable groups are defined by logFC thresholds only (logFC > 1, < -1, etc.). But classification should combine fold change with statistical significance (adjusted p-value). Also the UpSet plots are shown but numeric overlap metrics (counts per intersection) and Jaccard coefficients (Figure 3) should be accompanied by the sample sizes that generated them.
Response 5: The UpSet plots shown in Figure 3 show the intersection sizes – above the columns. The description of the calculation of the Jaccard coefficients is specified in the text of the publication.
Comments 6: At what circadian time were samples taken? Cold responses and translation are circadian-sensitive. State time of day and ensure all samples were harvested at the same circadian time.
Response 6: The experimental samples were collected at a fixed time – in the middle of the daylight hours (except for plants subjected to stress treatment, variants: Stress and Combo – middle of the daylight hours + 2 hours of temperature treatment). The corresponding corrections have been made to the text of the manuscript.
Comments 7: At least 30M reads per sample” is that for total RNA and for polysome libraries equally? Provide per-sample read counts in Supplementary Table.
Response 7: Added a corresponding table to the Supplementary materials section (Table A1.csv).
Comments 8: There is a wording slip: “The principal component distribution (Figure 1) shows that all experimental samples fall into two groups: samples reflecting transcriptional dynamics and transcriptional dynamics.” (lines 128–131) , likely you meant “transcriptional and translational samples” (fix wording).
Response 8: Make appropriate corrections.
Comments 9: (lines 187–230): Exactly how were polysome read counts normalized to total RNA counts? Did you compute TE for each gene as (polysome_TPM / total_TPM)? If so, how did you handle low counts? Did you use a statistical framework (Xtail, Babel, RiboDiff) to test significant TE changes? Normalization detail needed.
Response 9: To calculate TE we used edgeR, comparing poly vs. total.
Comments 10: WGCNA typically requires dozens of samples for robust module detection. Please state the number of samples used for WGCNA (transcriptional and translational networks). If sample N is small (e.g., < 12), treat WGCNA results cautiously and state that modules are exploratory.
Response 10: We used the same samples for WGCNA as for the previous analyses. Since there were only 2 replicates, we included a note that the modules found were exploratory and preliminary publication text.
Comments 11: Fix the double phrase in PCA section (“transcriptional dynamics and transcriptional dynamics” (lines ~128–131). There are a few other grammatical slips , run manuscript through careful editing.
Response 11: Make appropriate corrections.
Comments 12: I suggest to pick ~6 genes that show strong translational regulation but little transcriptional change, and validate by qPCR (total RNA) and polysome fraction qPCR or western blot (protein) to demonstrate true translational regulation (Section 2.6).
Response 12: We agree that the proposed additional qPCR and/or Western blot would undoubtedly increase the validity of the conclusions made in the publication. However, the experimental work was conducted 2 years ago, and we do not consider it acceptable to use samples stored for such a long period. At the same time, performing Western blot analysis would require significantly more than the 10 days allocated for the reviewer's response, and such analysis was not initially planned in the presented experiment, respectively, protein samples were not obtained. Nevertheless, we will definitely apply this approach in our research work in the future.
Comments 13: Also, rerun translational efficiency analyses using tools built for TE comparisons (Xtail, Babel, RiboDiff) and compare overlap with your current calls, present this in Supplementary.
Response 13: We are grateful for the proposed additional tools for quantitative translation analysis. However, we have a number of objections to this point:
-
Our study is not aimed at comparing bioinformatics tools for solving the above problem; benchmarking is a separate and rather labor-intensive area, such work can serve as a basis for a separate publication;
-
Two of the three proposed tools rely on edgeR (babel) and DESeq2 (Xtail), respectively, which probably suggests a classical approach based on differential expression analysis by these tools.
-
All three programs are primarily designed to analyze data obtained from ribosome profiling. In our case, polysomal profiling (undoubtedly, ribosome profiling has a higher resolution, and we plan to completely switch to it in the future).
Comments 14: For modules with clear hub genes (WGCNA), provide a short table of top 10 hub genes with annotations and discuss candidate regulators (TFs, RNA-binding proteins) that might mediate the level shift.
Response 14: Done, additional data added to Discussion section. Summary table for top-10 hub genes of each module added to Supplementary materials (Table A2.xlsx).
Reviewer 2 Report
Comments and Suggestions for Authors
Manuscript submitted to the International Journal of Molecular Science, MDPI, entitled:
Article
Dear authors,
The transcriptional and translational landscape of plant response to low temperature action
Aleksandra V. Suhorukova 1, Olga S. Pavlenko 1, Denis S.Sobolev 1, Ilya S. Demyanchuk 1, Valery N. Popov 1, and 4 Alexander A. Tyurin
Although the study has the potential to reach a broad audience and unusually presents transcriptional and translational results, the manuscript needs language revisions throughout.
For example, here is the corrected Abstract section.
Abstract: One of the unresolved questions in stress-response biology is how plants coordinate expression levels between the slow preparation phase and the rapid reactive response to stress exposure. In this work, we proposed a two-level analysis that examines both transcriptional and translational profiles of Solanum lycopersicum under short-term cold stress, hardening, and their combination. By combining polysome profiling and total transcriptome analysis, we revealed that expression under cold stress is not a simple linear process but a structurally distinct system with two coordinated regulation centers. Hardening triggers a strong transcriptional program focused on biogenesis, light signaling, and structural adaptations. In contrast, acute stress prompts selective translation of metabolic and defense proteins without prior transcriptional increase. Modular analysis (WGCNA) showed little overlap between transcriptional and translational networks, indicating functional differences between regulation levels. This work demonstrates that the cold response involves a strategic reallocation of resources between expression levels based on the type of signal. It bridges basic biology and applied breeding, providing targets promising for improving plant stress tolerance and advancing bioengineering of adaptive agriculture.
Title: Needs reformulation, this one is confusing, at least skip the word action on the end!
Keywords: Please do not use the exact wording as in the title.
Introduction:
Lines: 104-107: The goals of this study need to be formulated.
Results:
Lines 110-114: At the beginning of your results section, please identify the plant species you have been working with.
Figure 6. This diagram is confusing because the color code in the caption does not correspond with the color scheme shown in the picture.
Otherwise, this section is well-structured, and the results are interestingly described.
Discussion:
It is cumbersome to read, and it lacks a comparison with published data elsewhere. This section needs rewriting.
Material and Methods
Lines 529: Latin name must be written in Italian script!
Information about planting soil. The size of the used pots and the fertilization regime are missing, as well as the specification of the growth chamber used in the experiment. Please add all missing information.
Also used the correct unit for the light intensity.
Line 539: Missing citation for this experiment.
Line 547: Missing citation for this experiment.
Lines 553: Missing citation for this experiment. Correct expression in the mortar and pestle.
Line 556: Missing citation for this experiment.
All references for the suppliers' company need to be listed with the country of origin!
Sections 4.4 and 4.5 are missing a citation.
All references for the suppliers' companies need to be listed with the country of origin!
Conclusions
Too long and cumbersome to read, needs to be rewritten.
This manuscript requires correction in many areas before it can be considered for publication.
9.8.2025
Comments on the Quality of English LanguageThe manuscript needs language revisions throughout.
Author Response
We are grateful to the referee for reviewing our manuscript and providing constructive criticism. We have revised the language to be more clear and concise, and have included corrections as suggested by the reviewers to enhance the manuscript's clarity and informativeness.
Comments 1: Although the study has the potential to reach a broad audience and unusually presents transcriptional and translational results, the manuscript needs language revisions throughout.
Response 1: We conducted an additional round of collaborative editing of the manuscript text, taking into account suggestions and comments. We also revised the Abstract, Discussion and Conclusions.
Comments 2: Title: Needs reformulation, this one is confusing, at least skip the word action on the end!
Response 2: Reformulated the title
Comments 3: Keywords: Please do not use the exact wording as in the title.
Response 3: Have changed the list of keywords
Comments 4: Lines: 104-107: The goals of this study need to be formulated.
Response 4: The wording of the objectives of the presented work has been changed: The aim of this study was primarily to demonstrate significant functional differences between expression profiles at the transcriptional and translational stages in plants exposed to a stress factor (namely, low temperatures).
Comments 5: Lines 110-114: At the beginning of your results section, please identify the plant species you have been working with.
Response 5: The object of the study was indicated. It is also described in detail in the Materials and Methods section.
Comments 6: Figure 6. This diagram is confusing because the color code in the caption does not correspond with the color scheme shown in the picture.
Response 6: In this case, the colors indicated in the figure caption are the names of coexpressed gene modules, which are specified by the analysis program. We did not change the traditional approach to naming such modules. The color designations on the diagram itself are intended to distinguish coexpression modules identified at the transcription and translation stages.
Comments 7: Discussion: It is cumbersome to read, and it lacks a comparison with published data elsewhere. This section needs rewriting.
Response 7: We have carefully revised and shortened the Discussion section, and highlighted all changes in color in the text of the manuscript.
Comments 8: Lines 529: Latin name must be written in Italian script!
Response 8: Have corrected the spelling of the Latin name
Comments 9: Information about planting soil. The size of the used pots and the fertilization regime are missing, as well as the specification of the growth chamber used in the experiment. Please add all missing information.
Response 9: Added a detailed description of the cultivation conditions for model plants.
Comments 10: Also used the correct unit for the light intensity.
Response 10: Have corrected it.
Comments 11: Lines 539, 547, 553, 556: Missing citation for this experiment.
Response 11: Added previously missed citations and corrected terminology
Comments 12: All references for the suppliers' company need to be listed with the country of origin!
Response 12: The countries of origin of the reagents used were indicated.
Comments 13: Sections 4.4 and 4.5 are missing a citation.
Response 13: Removed section 4.4 as redundant.
Comments 14: Conclusions: Too long and cumbersome to read, needs to be rewritten.
Response 14: The conclusion section has been completely reworked. The edits have been highlighted in color in the text of the manuscript.
Reviewer 3 Report
Comments and Suggestions for Authors
The current manuscript addressed interesting questions: differences between transcription and translation under low temperature treatments in tomato plants.
The authors analyzed a large set of experimental data based on bulk RNA/protein analysis.
There are many points need to be improved.
Title: response is a process and need to be studied in dynamics. Here you study adaptation/reprogramming since you used only later time point (5d).
Line 9: “coordination of expression levels” = “coordination of gene expression levels”.
“slow preparation phase and the rapid reactive response to stress exposure” = response and adaptation.
Line 19: “strategic reallocation of resources between expression levels” – reallocation can be between organs, cell, but not between expression level.
Line 20: “forms” – built.
Line 33: “low distribution” ¿? Maybe low local?
Lines 39-62: Comprehensive description, however, with basial mistake. This description consider plant as homogenous cell population in which all cells respond with the same pathway. In the case of even leaf tissue, there are at least 7 cell types each respond by own trajectory. Mesophyll cell is 20-40 times large as epidermis, have a lytic (mainly) vacuole and this sub organelle is more sensitive to cold.
Therefore, it will be great to provide signaling as cascade of events in each cell types and coordination between cell types. Which cell type activate each signal etc.
Line 65: “gene expression regulation systems at the transcription and translation stages” with dynamical cellular resolution.
Line 68: “nature of the transcription” ???
Line 70: “another part of genes” ¿? Set of the genes.
Line 112: “Total RNA” – leaf? Root? Stem?
Figure 2: It is OK, but just difficult to read since description orientate vertically..
Moreover, here authors use DEG concept. Which threshold was settled ? 10%? 15%? 20%? Please, in future consider the in many case DEG is not biologically relevant since ignore “butterfly effect” and cell type specificity.
“The DEG (Differentially Expressed Genes) concept is fundamentally flawed — especially when applied to a single time point or to mutants from bulk RNA. Genes with small fold-changes may actually contribute more to adaptation than those with large changes. Moreover, in complex organ (as leaf) different cell types may have an opposite trend. (Recall the joystick: 100 grams of pressure can ultimately result in 100 kilograms of motor output.)”
Fig 3, 5: axis is too small and inviable if the same magnification as text were used.
Lines 197 – 211: rather M&M, do not contain results.
Line 399: “successful on p-value” ?? statistically significant.
Line 409: “mitotic cell size control checkpoint signalling,”??
Line 482: the response = the bulk adaptive response. Adaptaion mean establishing a new “balance” between different cell type since each cell type “run” own metabolic profile.
Line 565: which tissue?
Line 667: “different strategies for cell” – transcription and translation are not a strategy. Strategies is how plants choose cascades of events to activate under stress. It is not about “Long-term and rapid”…
Conclusion: can be shorter, less repetition from discussion.
Comments on the Quality of English LanguageLanguage should be more scientific.
Author Response
We are grateful to the referee for reviewing our manuscript and providing constructive criticism. We have revised the language to be more clear and concise, and have included corrections as suggested by the reviewers to enhance the manuscript's clarity and informativeness.
Comments 1: Title: response is a process and need to be studied in dynamics. Here you study adaptation/reprogramming since you used only later time point (5d).
Response 1: We have reformulated the title of the article to better reflect the essence and not mislead readers.
Comments 2: Line 9: “coordination of expression levels” = “coordination of gene expression levels”.
“slow preparation phase and the rapid reactive response to stress exposure” = response and adaptation.
Line 19: “strategic reallocation of resources between expression levels” – reallocation can be between organs, cell, but not between expression level.
Line 20: “forms” – built.
Response 2: We have also completely reworked the abstract taking into account your recommendations and suggestions from other reviewers. The main changes are highlighted in color in the text of the publication.
Comments 3: Line 33: “low distribution” ¿? Maybe low local?
Response 3: Corrected "low distribution" to "low".
Comments 4: Lines 39-62: Comprehensive description, however, with basial mistake. This description consider plant as homogenous cell population in which all cells respond with the same pathway. In the case of even leaf tissue, there are at least 7 cell types each respond by own trajectory. Mesophyll cell is 20-40 times large as epidermis, have a lytic (mainly) vacuole and this sub organelle is more sensitive to cold.
Therefore, it will be great to provide signaling as cascade of events in each cell types and coordination between cell types. Which cell type activate each signal etc.
Response 4: Yes, we completely agree. However, we did not have the opportunity to conduct research at the cellular level.
Comments 5: Line 65: “gene expression regulation systems at the transcription and translation stages” with dynamical cellular resolution.
Response 5: The corresponding clarification has been made.
Comments 6: Line 68: “nature of the transcription” ???
Response 6: The sentence has been reformulated. The correction has been highlighted in color in the text.
Comments 7: Line 70: “another part of genes” ¿? Set of the genes.
Response 7: Of course, a set of genes.
Comments 8: Line 112: “Total RNA” – leaf? Root? Stem?
Response 8: Only leaves were used for analysis.
Comments 9: Figure 2: It is OK, but just difficult to read since description orientate vertically.
Response 9: Agreed, there is a possibility that this can be corrected in the case of editorial layout. Also, in some applications for viewing pdf documents, you can temporarily flip the page using the key combination ctrl+left(right)arrow.
Comments 10: Moreover, here authors use DEG concept. Which threshold was settled ? 10%? 15%? 20%? Please, in future consider the in many case DEG is not biologically relevant since ignore “butterfly effect” and cell type specificity.
“The DEG (Differentially Expressed Genes) concept is fundamentally flawed — especially when applied to a single time point or to mutants from bulk RNA. Genes with small fold-changes may actually contribute more to adaptation than those with large changes. Moreover, in complex organ (as leaf) different cell types may have an opposite trend. (Recall the joystick: 100 grams of pressure can ultimately result in 100 kilograms of motor output.)”
Response 10: We fully agree that the use of the DEGs concept allows us to establish only general patterns and trends, some of which may be false positive (false negative) due to their statistical nature. However, to abandon this approach now means abandoning the work done and the data obtained (even if not entirely reliable).
Comments 11: Fig 3, 5: axis is too small and inviable if the same magnification as text were used.
Response 11: We have corrected figures 3 and 5. Due to low resolution of PNG files in manuscript we add additional folder with original figures to Supplementaru materials section.
Comments 12: Lines 197 – 211: rather M&M, do not contain results.
Response 12: We left the derivation of the formulas in the Results section because it may be easier for readers to view and discuss the indicator and its derivation in one place.
Comments 13: Line 399: “successful on p-value” ?? statistically significant.
Response 13: Of course, have corrected it.
Comments 14: Line 409: “mitotic cell size control checkpoint signalling,”??
Response 14: Yes, this is the content of the GO term -- https://amigo.geneontology.org/amigo/term/GO:0031567
Comments 15: Line 482: the response = the bulk adaptive response. Adaptaion mean establishing a new “balance” between different cell type since each cell type “run” own metabolic profile.
Response 15: The Discussion and Conclusion sections have been completely reworked, and corrections have been marked in color.
Comments 16: Line 565: which tissue?
Response 16: Leaves. This information is contained in section 4.1. We see no point in duplicating it multiple times.
Comments 17: Line 667: “different strategies for cell” – transcription and translation are not a strategy. Strategies is how plants choose cascades of events to activate under stress. It is not about “Long-term and rapid”…
Response 17: The Discussion and Conclusion sections have been completely reworked, and corrections have been marked in color.
Comments 18: Conclusion: can be shorter, less repetition from discussion.
Response 18: The Discussion and Conclusion sections have been completely reworked, and corrections have been marked in color.
Reviewer 4 Report
Comments and Suggestions for Authors
The submitted manuscript addresses the current issue of gene expression and translation in relation to the effects of low temperatures. The resistance of plants to low temperatures is highly relevant in practical terms, given the occurrence of spring frosts . After accepting the comments and notes on the manuscript, I believe that the manuscript will have a relatively high potential for citation after publication. It is somewhat unfortunate that the authors cite older literature. I believe that more current data is available at present. I recommend checking this. I recommend checking this. The text is carefully written, but I still recommend some additions or adjustments. The text includes the abbreviation AFC, which is not explained. Chapter 2 begins with a subchapter on experimental design, which should rather be part of the methodology. The methodology mentions the temperature, which is apparently related to the daily regime. So what was the temperature at night? Or did the temperatures not differ? This is not entirely clear from the text. The methodology should also mention the variety used. I believe that the light level is lower. Was it sufficient? In what medium were the plants grown? How was the experiment set up? Given the age of the plants, I believe that these are pot experiments. Figures 2 and 4 are included in the results. These are somewhat unclear. Please edit the discussion so that it is not descriptive.
Author Response
We are grateful to the referee for reviewing our manuscript and providing constructive criticism. We have revised the language to be more clear and concise, and have included corrections as suggested by the reviewers to enhance the manuscript's clarity and informativeness.
Comments 1: The text includes the abbreviation AFC, which is not explained.
Response 1: We've uncovered the abbreviation AFC -- ROS (reactive oxygen species).
Comments 2: Chapter 2 begins with a subchapter on experimental design, which should rather be part of the methodology.
Response 2: Moved this subsection to the Materials and Methods section.
Comments 3: The methodology mentions the temperature, which is apparently related to the daily regime. So what was the temperature at night? Or did the temperatures not differ? This is not entirely clear from the text. The methodology should also mention the variety used.
Response 3: The temperature was kept constant, this was reflected in the edited version of the article text.
Comments 4: I believe that the light level is lower. Was it sufficient?
Response 4: We have corrected the values in the text. The correct value is 200 µmol/(m2s) (approximately 10 klux).
Comments 5: In what medium were the plants grown? How was the experiment set up? Given the age of the plants, I believe that these are pot experiments.
Response 5: Yes, the plants were grown in soil, this was clearly indicated in the corrected text of the article.
Comments 6: Please edit the discussion so that it is not descriptive.
Response 6: Completely reworked the text of the Discussion section. All errors are highlighted in color.
Round 2
Reviewer 1 Report
Comments and Suggestions for Authors
The authors have addressed my concerns adequately, and hence i recommend the publication of this manuscript in its current form.
Author Response
We are grateful to the reviewer for re-reading our manuscript aimed at improving the presented work.
Reviewer 2 Report
Comments and Suggestions for Authors
The authors of this manuscript corrected and revised the original manuscript into the current version, which can be considered for publication.
28.8.2025
Author Response

(The authors gave the same response as above.)

Reviewer 3 Report
Comments and Suggestions for Authors
Thank you, for response.
Please, take into account these poit as well:
Line 10: "of Solanum lycopersicum leaf under"..
Line 18: "between expression levels " may be confusing as may mean gene expression level. Maybe more precise: between transcription and translation?
Lin 188: "Based on a previously proposed methodology [13,14], we 188
applied normalisation to the starting transcript quantities for comparisons between 189
samples, which requires analysis of total transcriptomes. " Please, reformulate more clearly.
Line 236: "The next stage of the presented study was " redundat for scientific paper.
Line 334: "the genetic response " . what is genetic response form biologicalpoint of view? You did not study mutants.
Line 479: positive regulation of signalling,- which signalling? Signalling mean a kinetic_ form which cell to which cell/organell?
Please, it is crucial to add to discussion at least short part about limitation as ignorationof butterfly effect, cell type specificity and, if possible, your "speculation" abot biolocal cascade.
Line 585: Total RNA was isolated = Total RNA was isolated from the leaf (young/older??)
Line 631: Identification of genes with altered expression levels - pease, add threshold: 5%-30% under which gene expression consider as different.
Author Response
We are grateful to the reviewer for re-reading our manuscript and for constructive criticism aimed at improving the presented work.
Comments 1: Line 10: "of Solanum lycopersicum leaf under"..
Response 1: We have reformulated this sentence. The changes have been highlighted in color in the text of the manuscript.
Comments 2: Line 18: "between expression levels " may be confusing as may mean gene expression level. Maybe more precise: between transcription and translation?
Response 2: Agreed. We used the suggested wording.
Comments 3: Lin 188: "Based on a previously proposed methodology [13,14], we 188
applied normalisation to the starting transcript quantities for comparisons between 189
samples, which requires analysis of total transcriptomes. " Please, reformulate more clearly.
Response 3: The sentence was reformulated as follows:“Since the number of translated transcripts largely depends on the initial number of mRNA molecules of a given type, normalization of poly/total is required – the so-called translational ratio – how effectively molecules of a particular mRNA type enter the translation process.”. The changes are highlighted in the text in color.
Comments 4: Line 236: "The next stage of the presented study was " redundat for scientific paper.
Response 4: This sentence has been completely removed from the text.
Comments 5: Line 334: "the genetic response " . what is genetic response form biologicalpoint of view? You did not study mutants.
Response 5: We completely agree. The changes made are highlighted in color.
Comments 6: Line 479: positive regulation of signalling,- which signalling? Signalling mean a kinetic_ form which cell to which cell/organell?
Response 6: This is the content of the term GO:0023056 (Any process that activates, maintains or increases the frequency, rate or extent of a signaling process.") We have added a clarification to the text that these results are based on GO-enrichment analysis.
Comments 7: Please, it is crucial to add to discussion at least short part about limitation as ignorationof butterfly effect, cell type specificity and, if possible, your "speculation" abot biolocal cascade.
Response 7: Added the corresponding paragraph to the Discussion section.
Comments 8: Line 585: Total RNA was isolated = Total RNA was isolated from the leaf (young/older??)
Response 8: We made the necessary edits.
Comments 9: Line 631: Identification of genes with altered expression levels - pease, add threshold: 5%-30% under which gene expression consider as different.
Response 9: As a threshold for determining whether genes are differentially expressed, we used the logFC parameter, the actual threshold value being |logFC|= 1.
Round 3
Reviewer 3 Report
Comments and Suggestions for Authors
Thank you!
Please, do minor corrections.
Fig 3 and 5: numbers in black box is too small. Please, increase font size, made it bold or so.
Line 549: "butterfly effcet" mean something different: small changes in gene exoression in one cell types (stomat for exmaple) may lead to significant chnages in the other cell type and even total cell re-programming far from initial point. This is a key problem in total DEG and even in single cell transcriptomics concepts.
My best regards!
Author Response
The authors are grateful to the reviewer for the attention paid to our manuscript and assistance in its iterative improvement.
Comments 1: Fig 3 and 5: numbers in black box is too small. Please, increase font size, made it bold or so.
Response 1: Made the numbers larger.
Comments 2: Line 549: "butterfly effcet" mean something different: small changes in gene exoression in one cell types (stomat for exmaple) may lead to significant chnages in the other cell type and even total cell re-programming far from initial point. This is a key problem in total DEG and even in single cell transcriptomics concepts.
Response 2: Made some minor clarifications to the text.